# Reward Adaptation via Q-Manipulation

## Abstract

In this paper, we introduce reward adaptation (RA), the problem where the learning agent adapts to a target reward function based on one or multiple existing behaviors learned a priori based on their corresponding source reward functions, providing a new perspective of modular reinforcement learning. Reward adaptation has many applications, such as adapting an autonomous driving agent that can already operate either fast or safe to operating both fast and safe. Learning the target behavior from scratch is possible but inefficient given the source behaviors available. Assuming that the target reward function is a polynomial function of the source reward functions, we propose an approach to reward adaptation by manipulating variants of the Q function for the source behaviors, which are assumed to be accessible and obtained when learning the source behaviors prior to learning the target behavior. It results in a novel method named "Q-Manipulation" that enables action pruning before learning the target. We formally prove that our pruning strategy for improving sample complexity does not affect the optimality of the returned policy. Comparison with baselines is performed in a variety of synthetic and simulation domains to demonstrate its effectiveness and generalizability.

## 1 Introduction

Reinforcement Learning (RL) Watkins (1989); Sutton & Barto (2018) represents a class of learning methods that allow agents to learn from interacting with the environment. RL has demonstrated great successes in simulation domains such as Atari games Mnih et al. (2015), MuJoCo, board games (chess Campbell et al. (2002), Go Silver et al. (2016)), etc. However, applying RL to real-world problems is still challenging since the agents are often required to interact with the physical world to learn, which can be expensive.

The key to address such a problem is to reduce (online) sample complexity, which can often be achieved in one of the following ways in RL: 1) learning optimization, 2) transfer learning (including domain adaptation), 3) model reuse, and, more recently, 4) offline RL. In this paper, we introduce Reward Adaptation (RA), the problem where the learning agent adapts to a target reward function given one or multiple existing behaviors learned a priori with their corresponding source reward functions. RA has many useful applications, such as adapting an autonomous driving agent to drive fast and safe when it already knows how to operate either fast or safe. The core idea of our approach to RA is to leverage "knowledge" about the existing behaviors to learn the target behavior. From this aspect, our approach bears similarities with model reuse, and contributes a unique perspective to modular RL. As a result, our approach can benefit from a repository of source behaviors to create new and potentially more complex target behaviors. In this work, we restrict to behaviors with the same discrete state and action spaces but different reward functions. RA may also be viewed a special form of transfer learning where the knowledge from the source domains is used to expedite learning in the target domain, although not so directly as in transfer learning. Furthermore, in contrast to domain adaptation Peng et al. (2018); Eysenbach et al. (2020) in transfer learning, we assume access to the target domain during learning and are given only learned behaviors in the source domains (i.e., no access to the source domains while learning the target).

To better conceptualize the RA problem, consider a grid-world as shown in Fig. 1, which is an expansion of the Dollar-Euro domain described in Russell & Zimdars (2003). In this domain, the agent can move to any of its adjacent locations at any step. The agent's initial location is colored in yellow and the terminal locations are colored pink or green, which correspond to the two different reward functions (i.e., collecting dollars or euros), respectively. The terminal lo-

cations with a single color return a reward of $1.0$ in their reward functions, respectively, and the terminal location with split colors returns a reward of $0.6$ under both reward functions. In RA, we assume that the optimal behavior under each such reward function is given, referred to as a source behavior. An example of a target domain is characterized by a target reward function that considers both dollars and euros. Although the problem setting resembles that of Q-Decomposition Russell & Zimdars (2003), the focus of Q-Decomposition is to learn under each source reward function independently while constructing the target behavior. In contrast, RA is focused on how to leverage the source behaviors that are available while learning the target behavior.

Learning the target behavior from scratch is possible but inefficient. Instead, we propose an approach to reward adaptation named "Q-Manipulation". We assume that the learning agent maintains two variants of the Q function (referred to as Q and Q-min) for each source behavior: they are computed a priori when the source behavior is learned. We note that all RL methods using value function estimates in learning have access to Q and, with minor modifications, to Q-min as well (more details later). We further assume that the target reward function is a polynomial function of the source reward functions. The Q and Q-min's under the source reward functions are used to compute an upper and lower bounds

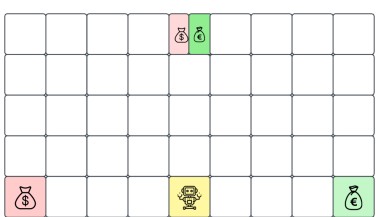

Figure 1: The Dollar-Euro domain.

of the expected return under the target reward function. We then tighten the bounds using reward shaping to maximize pruning opportunities. Such a process enables us to prune a substantial number of actions before learning the target without affecting the optimality of the returned policy, leading to guaranteed improvements in sample complexity. We remark that even though computing and maintaining the Q and Q-min for each source behavior requires additional resources, its benefits largely outweigh the costs for practical applications, especially in situations where accessing the target domain is expensive (e.g., when the rewards in the target domain are provided by users).

Our core contributions are: We introduce the problem of reward adaptation. To the best of our knowledge, other than approaches that compose complex behaviors from existing behaviors, such as in hierarchical and composable RL Simpkins & Isbell (2019); Doroodgar & Nejat (2010) (see related work), little effort has been invested to exploit these available behaviors. Thus, our work contributes a unique perspective to modular RL. Reward adaptation may also be viewed as a special form of transfer learning with a focus on adapting the reward function. We propose Q-Manipulation for reward adaptation to expedite learning the target behavior given the source behaviors. It reuses knowledge from the source behaviors in an indirect way during learning. We prove that our approach does not affect the optimality of the returned policy and thus guarantees improvements in sample complexity. Our evaluations provide further insights into the effectiveness of Q-Manipulation with respect to several baselines and its ability to generalize to various reward adaptation settings.

## 2 METHODOLOGY

First, we briefly introduce the problem setting of reinforcement learning (RL) before defining reward adaptation (RA). In this work, we assume discrete state and action spaces. Extension to continuous state and action spaces is discussed in Sec. 5 and will be addressed in our future work. In RL, the task environment is modeled as an MDP $M = (S, A, T, R, \gamma)$, where $S$ is the state space, $A$ is the action space, $T : S \times A \times S \to [0, 1]$ is the transition function, $R : S \times A \to \mathbb{R}$ is the reward function, and $\gamma$ is the discount factor. At every step $t$, the RL agent observes state $s_t$ and takes an action $a_t \in A$. As a result, the agent progresses to state $s_{t+1}$ with probability $T(s_t, a_t, s_{t+1})$, and receives a reward $r_t$. The goal is to search for a policy that maximizes the expected cumulative reward. We use $\pi$ to denote a policy as a mapping from $S$ to $A$. The Q function of the optimal policy $\pi^*$ is defined by:

$$Q_R^*(s, a) = \max_\pi \left[ \mathbb{E} \left[ \sum_{t=0}^\infty \gamma^t r_t | s_0 = s, a_0 = a, \pi \right] \right] \tag{1}$$

Our solution to RA is related to a variant of the Q function, which we refer to as Q-min (note the $\min$ operator below), denoted by $Q_R^\mu$:

$$Q_R^\mu(s, a) = \min_\pi \left[ \mathbb{E} \left[ \sum_{t=0}^\infty \gamma^t r_t | s_0 = s, a_0 = a, \pi \right] \right] \tag{2}$$

Intuitively, $Q_R^\mu$ above represents the Q function of the policy that leads to the minimum expected return, in contrast to $Q_R^*$ that corresponds to the maximum expected return. Furthermore, the following lemma establishes a connection between $Q_R^\mu$ and a max form of Q. Throughout the paper, proofs, if omitted, are included in the appendix.

**Lemma 1.**

$$Q_R^\mu(s, a) = -Q_{-R}^*(s, a) \tag{3}$$

*where $Q_{-R}^*(s, a)$ denotes the Q function of the optimal policy in Eq. 1 under $-R$.*

## 2.1 REWARD ADAPTATION

**Definition 1** (Reward Adaptation (RA)). *Given a set of source behaviors trained under an MDP $M \setminus R = (S, A, T, \cdot, \gamma)$ with different source reward functions, compute the optimal policy for a target reward function under $M \setminus R$.*

In Def. 1 above, note that we assume the same state and action spaces for the source and target behaviors. To derive a solution to RA while utilizing the information encoded in the source behaviors, we propose Q-Manipulation, an action pruning strategy that ensures that only unnecessary actions are pruned. To achieve this, we aim to compute an upper and lower bounds of the expected return under the target reward function based solely on information from the source behaviors. Intuitively, if the lower bound of an action $a$ is higher than the upper bound of action $\widehat{a}$ under a state $s$, $\widehat{a}$ can be pruned. In Q-Manipulation, we derive these bounds based on manipulating variants of the Q function for the source behaviors to maximize pruning opportunities.

In particular, we assume that the agent obtains both Q and Q-min (or equivalently $Q_R^*$ and $Q_{-R}^*$) when learning the corresponding source behavior and maintains them for future use. Next, we will show how to derive an upper and lower bound of the expected return in the target domain using these Q's from the source domains given a general relationship known between the reward functions.

**Definition 2** (Reward Adaptation with Q Variants). *Given a reward adaptation problem where $Q_{R_i}^*$ and $Q_{-R_i}^*$ are accessible for each source domain indexed by $i$, compute the optimal policy under a target reward function that is a polynomial function of the source reward functions:*

$$\mathcal{R} = \sum_{i_1+i_2+i_3+\ldots i_n \leq m} a_{i_1 i_2 \ldots i_n} R_1^{i_1} R_2^{i_2} \cdots R_n^{i_n} \tag{4}$$

*where $R_i$ is the source reward function for the $i^{th}$ domain.*

We facilitate a formal treatment of RA by focusing on cases when the relationship in Eq. 4 is exact. In our evaluation, however, we analyze the more general case when the available source reward functions can only be used to approximate the target reward function.

## 2.2 DERIVING UPPER AND LOWER BOUNDS UNDER $\mathcal{R}$

To derive bounds under $\mathcal{R}$, we start with simpler forms of Eq. 4 and then combine the results. In the following, we denote the upper bound as $\mathbb{Q}_\mathcal{R}^*$ and the lower bound as $\mathbb{Q}_\mathcal{R}^\mu$. We refer to a reward function as positive when all rewards are non-negative. We also assume in the following that the influence of discounting can be safely ignored, e.g., when MDPs with absorbing states are considered.

**Lemma 2.** *When $\mathcal{R} = R^m$ and $R$ is positive, an upper and lower bounds of the expected return under $\mathcal{R}$ are given, respectively, by:*

$$\mathbb{Q}_\mathcal{R}^* = {Q_R^*}^m \geq Q_{R^m}^* = Q_\mathcal{R}^* \tag{5}$$

$$\mathbb{Q}_\mathcal{R}^\mu = -|Q_{-R}^*|^m \leq -Q_{-R^m}^* = Q_\mathcal{R}^\mu \tag{6}$$

**Lemma 3.** *When $\mathcal{R} = R_i \times R_j$, and both $R_i$ and $R_j$ are positive, an upper and lower bounds of the expected return under $\mathcal{R}$ is given, respectively, by:*

$$\mathbb{Q}_\mathcal{R}^* = Q_{R_i}^* \times Q_{R_j}^* \geq Q_{R_i R_j}^* = Q_\mathcal{R}^* \tag{7}$$

$$\mathbb{Q}_\mathcal{R}^\mu = -|Q_{-R_i}^*| \times |Q_{-R_j}^*| \leq -Q_{-R_i R_j}^* = Q_\mathcal{R}^\mu \tag{8}$$

**Lemma 4.** *When $\mathcal{R} = aR_i + bR_j$ $(a > 0, b > 0)$, an upper and lower bounds of the expected return under $\mathcal{R}$ are given, respectively, by:*

$$\mathbb{Q}_{\mathcal{R}}^* = aQ_{R_i}^* + bQ_{R_j}^* \geq Q_{aR_i+bR_j}^* = Q_{\mathcal{R}}^* \tag{9}$$

$$\mathbb{Q}_{\mathcal{R}}^\mu = -(aQ_{-R_i}^* + bQ_{-R_j}^*) \leq -Q_{-(aR_i+bR_j)}^* = Q_{\mathcal{R}}^\mu \tag{10}$$

**Theorem 1.** *Given $\mathcal{R}$ in the form of Eq. 4 where every $a_{i_1 i_2 \ldots i_n} > 0$ and every source reward function is positive, an upper and lower bounds of the expected return under $\mathcal{R}$ are given, respectively, by:*

$$\mathbb{Q}_{\mathcal{R}}^* = \sum_{i_1+i_2+i_3+\ldots i_n \leq m} a_{i_1 i_2 \ldots i_n} |Q_{R_1}^*|^{i_1} \times |Q_{R_2}^*|^{i_2} \times \ldots \times |Q_{R_n}^*|^{i_n} \geq Q_{\mathcal{R}}^* \tag{11}$$

$$\mathbb{Q}_{\mathcal{R}}^\mu = -\left[ \sum_{i_1+i_2+i_3+\ldots i_n \leq m} a_{i_1 i_2 \ldots i_n} |Q_{-R_1}^*|^{i_1} \times |Q_{-R_2}^*|^{i_2} \times \ldots \times |Q_{-R_n}^*|^{i_n} \right] \leq Q_{\mathcal{R}}^\mu \tag{12}$$

The upper and lower bounds above are tight in the sense that the equalities may hold. Note that every source reward function is assumed to be positive in Theorem 1. We evaluated the robustness of our approach in the appendix when such an assumption does not hold. It is worth noting that, when $\mathcal{R}$ is simply a linearly weighted sum of the source reward functions, this assumption is no longer required and all the absolute operators in Theorem 1 may be removed (see Lemma 4). This allows our approach to apply to any source reward functions when $\mathcal{R}$ assumes a linear form.

## 2.3 REWARD SHAPING FOR "TIGHTENING" THE BOUNDS

Intuitively, the smaller the distance between the bounds is, the more pruning opportunities there may be. Even though we cannot tighten the bounds without additional information, in the following, we show that we can "effectively" achieve the same effect by reward shaping Ng et al. (1999). Reward shaping has been widely used to support more informative reward functions to guide learning. In our case, however, we apply it with the objective to reduce the distance between the computed bounds.

A shaping function of a reward function $R$ has the form of $\mathcal{F} = \gamma * \Phi(s') - \Phi(s)$ following Ng et al. (1999), where $\Phi$ is referred to as a potential function. The shaped reward function, denoted by $R_F$, satisfies $R_F = R + \mathcal{F}$. Applying reward shaping this way does not affect the optimal policy, i.e., the optimal policy remains invariant. Next, we first establish the effects of applying reward shaping to the target reward function $\mathcal{R}$ on $Q_{\mathcal{R}}^*$ and $Q_{\mathcal{R}}^\mu$ below.

**Lemma 5.** *$Q_{\mathcal{R}}^*$ and $Q_{\mathcal{R}}^\mu$ after applying reward shaping to $\mathcal{R}$ with the shaping function $\mathcal{F}$ are given, respectively, by:*

$$Q_{\mathcal{R}_{\mathcal{F}}}^*(s, a) = Q_{\mathcal{R}}^*(s, a) - \Phi(s) \tag{13}$$

$$Q_{\mathcal{R}_{\mathcal{F}}}^\mu(s, a) = Q_{\mathcal{R}}^\mu(s, a) + \Phi(s) \tag{14}$$

*Proof:*

$$Q_{\mathcal{R}_{\mathcal{F}}}^*(s, a) = Q_{\mathcal{R}}^*(s, a) - \Phi(s) \quad [\textit{Ng et al. (1999)}]$$

$$Q_{\mathcal{R}_{\mathcal{F}}}^\mu(s, a) = -\left[ Q_{-\mathcal{R}_{\mathcal{F}}}^*(s, a) \right] \quad [\textit{Lemma 1}]$$

$$= -\left[ Q_{-\mathcal{R}}^*(s, a) - \Phi(s) \right]$$

$$= -Q_{-\mathcal{R}}^*(s, a) + \Phi(s)$$

$$= Q_{\mathcal{R}}^\mu(s, a) + \Phi(s)$$

It is crucial to observe that reward shaping updates $Q_{\mathcal{R}}^*$ and $Q_{\mathcal{R}}^\mu$ in the opposite directions. Since the upper and lower bounds in Theorem 1 will be influenced in the same way by reward shaping, it can be used to reduce the distance between the bounds, effectively tightening them for more pruning opportunities.

To compute the shaping function, we apply a linear programming formulation. More specifically, we minimize the sum of distances between the bounds after shaping, subject to the constraint that the

shaped upper bound remains greater than or equal to the shaped lower bound. This constraint ensures that the distances between the bounds are always positive. More formally,

$$
\begin{aligned}
&\min_{\Phi(s)} \sum_{s,a} \mathbb{Q}_{\mathcal{R}}^{*}(s,a) - \mathbb{Q}_{\mathcal{R}}^{\mu}(s,a) - 2 * \Phi(s) \\
&\text{s.t.} \quad \forall s \in S, \forall a \in A \\
&\mathbb{Q}_{\mathcal{R}}^{*}(s,a) - \Phi(s) \geq \mathbb{Q}_{\mathcal{R}}^{\mu}(s,a) + \Phi(s)
\end{aligned}
\tag{15}
$$

Note that the optimization above does not directly maximize the pruning opportunities (which is more difficult). In Q-Manipulation, we use the shaped bounds for action pruning and only use the remaining set of actions under each state when learning the target.

**Theorem 2** (optimality). *Given a problem of reward adaptation with Q variants (Def. 2), the optimal policies under $\mathcal{R}$ remain invariant under Q-Manipulation.*

*Proof.* Let

$$
\begin{aligned}
A_p(s) &= \{\widehat{a} | \; \exists a \; \mathbb{Q}_{\mathcal{R}_{\mathcal{F}}}^{\mu}(s,a) > \mathbb{Q}_{\mathcal{R}_{\mathcal{F}}}^{*}(s,\widehat{a}); a \neq \widehat{a}\} \\
\tilde{A}(s) &= A(s) \setminus A_p(s)
\end{aligned}
$$

where $A_p(s)$ represents the set of pruned actions under set $s$ and $\tilde{A}$ represents the remaining set of actions. To retain all optimal policies, it must be satisfied that none of the optimal actions under each state are pruned. Assuming that a pruned action $\widehat{a}$ under $s$ is an optimal action, we must have

$$
\forall a \; Q_{\mathcal{R}_{\mathcal{F}}}^{*}(s,a) \leq Q_{\mathcal{R}_{\mathcal{F}}}^{*}(s,\widehat{a})
$$

Given that Q-Manipulation only prunes an action $\widehat{a}$ under $s$ when $\exists a \; \mathbb{Q}_{\mathcal{R}_{\mathcal{F}}}^{\mu}(s,a) > \mathbb{Q}_{\mathcal{R}_{\mathcal{F}}}^{*}(s,\widehat{a})$, we can derive that

$$
Q_{\mathcal{R}_{\mathcal{F}}}^{\mu}(s,a) \geq \mathbb{Q}_{\mathcal{R}_{\mathcal{F}}}^{\mu}(s,a) > \mathbb{Q}_{\mathcal{R}_{\mathcal{F}}}^{*}(s,\widehat{a}) \geq Q_{\mathcal{R}_{\mathcal{F}}}^{*}(s,\widehat{a}) \geq Q_{\mathcal{R}_{\mathcal{F}}}^{*}(s,a),
$$

resulting in a contradiction that

$$
Q_{\mathcal{R}_{\mathcal{F}}}^{\mu}(s,a) > Q_{\mathcal{R}_{\mathcal{F}}}^{*}(s,a)
$$

As a result, we know that all optimal actions and hence policies are retained. □

## 3 EVALUATION

In this section, the primary objectives include analyzing the performance of Q-Manipulation (Q-M) and substantiating the claimed benefits of Q-M for learning the target behavior. Since the focus here is on sample complexity, we compare Q-M with several baselines under a simple learning framework, the basic temporal difference learning in discrete domains Sutton & Barto (2018). The baselines chosen are those that share certain characteristics with our Q-M implementation (running Q-learning). In particular, we compare Q-M with the traditional Q-learning (Q), Q-Decomposition (Q-D) that leverages reward decomposition during learning Russell & Zimdars (2003), and Q-learning with automatic reward shaping Marthi (2007) (R-S). For R-S, the shaping function is given by the value function of an abstract MDP that is either automatically (for synthetic domains) or manually crafted (for simulation domains) from the original MDP. K-means clustering was used to auto-generate abstract MDPs with $k = |S|/1.4$ based on their adjacency matrices.

We evaluated with various simulation and synthetic domains to validate the effectiveness and generalizability of Q-M. Additional results are reported in the appendix. For all evaluations, we averaged over 20 runs. The learning rate was set at 0.1 with $\epsilon$-greedy exploration where $\epsilon$ was decayed over time. For Q-M, we pre-train the source behaviors to obtain both $Q_{R}^{*}$ and $Q_{-R}^{*}$. Even though Q-M has access to additional information, we do not consider the extra costs (for learning $Q_{-R}^{*}$ or equivalently Q-min) since they are assumed to be incurred before considering the task in hand (i.e., learning the target behavior) and inexpensive to obtain when learning the source behaviors. See Sec. 5 for more discussion. Note also that since Q-D only works in domains where the target reward function is a linear sum of the source reward functions, it is missing in some of our evaluations and comparisons. Next, we briefly describe the domains.

### 3.1 DOMAIN DESCRIPTION

**Dollar-Euro:** A 45 states and 4 actions grid-world domain as illustrated in Fig. 1. **Source Domain 1 with $R_1$ (collecting dollars):** The agent obtains a reward of 1.0 for reaching the location labeled with "$", and 0.6 for reaching the location labeled with both $ and €. **Source Domain 2 with $R_2$ (collecting euros):** The agent obtains a reward of 1.0 for reaching the location labeled with €, and 0.6 for reaching the location labeled with both $ and €. **Target Domain with $\mathcal{R}$:** In the original domain, $\mathcal{R} = R_1 + R_2$. We also study a case when $\mathcal{R} = R_1^4 + R_2^3$. The living reward is 0.

**Frozen Lake:** A standard toy-text gym environment with 16 states and 4 actions. An episode terminates when the agent falls into the frozen lake or reaches the goal. **Source Domain 1 with $R_1$ (seeking goal):** The agent is rewarded 10 for reaching the goal. **Source Domain 2 with $R_2$ (avoiding falling):** The agent is penalized by $-1$ for falling into the lake. **Target Domain with $\mathcal{R}$:** Avoid the frozen lake and reach the goal location, $\mathcal{R} = R_1 + R_2$. The living reward is 0.

**Race Track:** A 49 states and 9 actions grid-world domain. The 9 actions correspond to adding $-1$, 0, or $+1$ to the current velocity while going forward, turning left, or turning right. An initial location, a goal location, and obstacles make up the race track. An episode ends when the agent reaches the goal position, crashes, or exhausts the total number of steps. **Source Domain 1 with $R_1$ (avoiding obstacles):** The agent obtains a negative reward of $-10$ for collision and otherwise 0. **Source Domain 2 with $R_2$ (seeking goal):** The agent obtains a reward of 100 for reaching the goal and otherwise 0. **Source Domain 3 with $R_3$ (minimizing running steps):** The agent obtains a negative reward of $-1$ for each step. **Target Domain with $\mathcal{R}$:** Reach the goal in the least number of steps while avoiding all obstacles: $\mathcal{R} = R_1 + R_2 + R_3$.

**Auto-Generated Domains:** To analyze how Q-M scales with increasing state and action spaces, these domains are constructed by randomly generating MDPs. Each MDP has an action space size between 4 and 20, a state space size between $|A|$ and 100, a transition matrix, and a terminal state. The reward functions of the source behaviors are also randomly generated. For each source domain, we randomly select a set of states of size between $[0, |S|]$, each state with a negative reward randomly generated between [-20, 0]; the remaining states have a reward of 0. When $\mathcal{R}$ is a non-linear function of the source reward functions, we generate positive rewards randomly between [0, 20]. Reaching the terminal state gives a positive reward randomly generated from [0, 100]. We generated 6 domains under these settings and studied various relationships between the target and source reward functions. Larger domains were generated similarly for additional results reported in the appendix.

### 3.2 RESULTS

Results summarizing all the domains presented here, actions pruned by Q-M, and time performances for the different methods are presented in Tab. 1. We can see that Q-M pruned out a substantial number of actions under all domains evaluated, ranging from 7.8% to 42.7% of the original set of actions. The pruning efficiency does not seem to be directly correlated with the form of $\mathcal{R}$ or the size of the domain, which we will investigate more in future work. Q-M is comparable with the baselines in terms of time performance even with the computation time for the linear programming incorporated. These results validated Q-M as an effective methodology for pruning unnecessary actions before learning.

To analyze the effects of action pruning, we compared the convergence of the different methods. The results for when $\mathcal{R}$ is a linear function of the source functions are presented in Figs. 2 (simulation) and 3 (synthetic). We can see that action pruning expedited convergence such that Q-M outperformed the baselines significantly in almost all domains evaluated. We also note that the convergence rate was observed to be directly related to the pruning efficiency, which aligns with our intuition. For example, only 7.8% of actions were pruned in Frozen Lake, resulting in the smallest improvement among all domains. To better see this, we also visualized the percentages of actions pruned under each state for the three simulation domains (see Fig. 2). Intuitively, actions pruned from different states may have different significance for convergence. For Q-M, the heat maps reveals that action pruning occurred more around the initial and goal states. We will further analyze this in future work.

Convergence comparisons for domains where $\mathcal{R}$ assumes a non-linear form are presented in Fig. 4, with an action pruning heat map for the only simulation domain evaluated with a non-linear $\mathcal{R}$. Similar results were observed.

| Environment | $\mathcal{R}$ | $|S|$ | $|A|$ | Actions Pruned | Time (s) Q | Q-D | Q-M | R-S |
|---|---|---|---|---|---|---|---|---|
| Dollar Euro | $R_1 + R_2$ | 45 | 4 | 32 | 0.1 | 0.1 | 0.23 | 0.1 |
| Dollar Euro | $R_1^4 + R_2^3$ | 45 | 4 | 28 | 28.3 | - | 30.0 | 28.29 |
| Frozen Lake | $R_1 + R_2$ | 16 | 4 | 5 | 3.8 | 3.89 | 3.7 | 4.1 |
| Race track | $R_1 + R_2 + R_3$ | 49 | 9 | 51 | 75.58 | 41.64 | 77.13 | 53.34 |
| Auto-gen 1 | $R_1 + R_2$ | 26 | 4 | 35 | 0.92 | 1.11 | 0.88 | 0.97 |
| Auto-gen 2 | $R_1 + R_2$ | 57 | 12 | 167 | 1.97 | 3.5 | 2.27 | 2.15 |
| Auto-gen 3 | $R_1 + R_2$ | 56 | 14 | 242 | 1.66 | 3.7 | 1.73 | 1.92 |
| Auto-gen 4 | $R_1 + R_2 + R_3 + R_4$ | 99 | 18 | 247 | 1.5 | 5.6 | 1.6 | 3.09 |
| Auto-gen 5 | $R_1^3 + R_2^3$ | 70 | 4 | 34 | 1.45 | - | 2.7 | 1.79 |
| Auto-gen 6 | $R_1 \times R_2 + R_3 + R_4$ | 80 | 17 | 581 | 1.49 | - | 1.41 | 2.04 |

Table 1: Summary of domains, actions pruned by Q-M, and time performance comparisons. For Q-M, the reported times include that for computing the shaping reward function using linear programming.

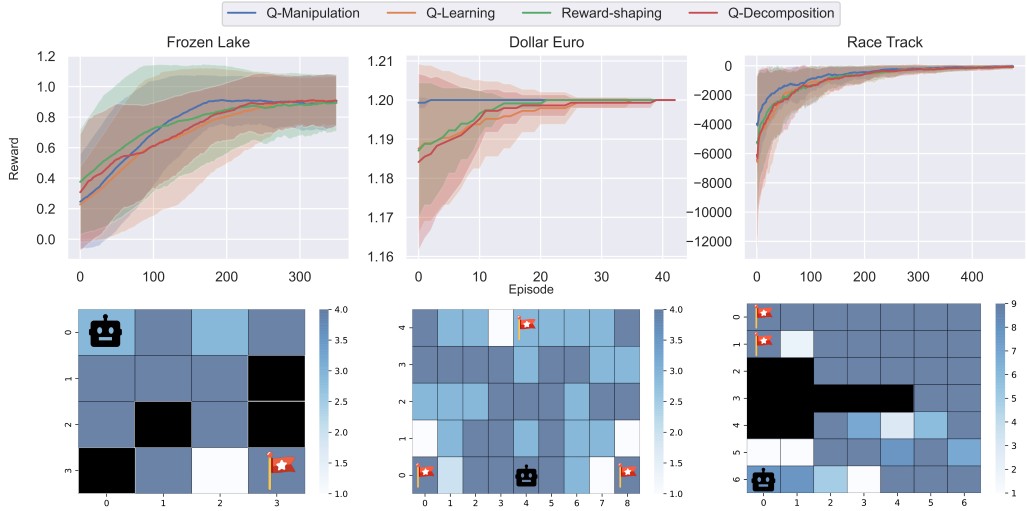

Figure 2: Convergence comparisons (top row) and action-pruning heat maps (bottom row) for the simulation domains where $\mathcal{R}$ has a linear form. In the heat maps, the lighter blue the location is, the smaller the set of actions is left. The flag symbols represent goal locations, agent symbols represent the initial locations, and locations blocked by obstacles are colored black.

We also evaluated the performance of Q-M when the target reward function can only be approximated by the source reward functions. The domain is an auto-generated MDP. In the base case, $\mathcal{R} = R_1 + R_2$. We then add different levels of Gaussian noise to $\mathcal{R}$ to create new target domains where Q-M is only allowed to approximate the new target reward function using $R_1$ and $R_2$ via linear regression. The level of noise as the standard deviation is set in proportion to the original $\mathcal{R}$'s mean range for generation (i.e., 10). It is important to note that the theoretical guarantee of optimality is lost in such an evaluation setting. We compared Q-M (given the approximate target function), shown as solid lines, with Q learning (given the true target function), shown as dashed lines, in each case in Fig. 5. Q-M started to deviate from the optimal solution substantially at 4% noise level. One of the problems here is the limitation of linear regression. We will study the robustness of Q-M in future.

## 4 RELATED WORK

**Reward and Q-Decomposition**: Reward structure can significantly influence the effectiveness of an RL agent Silver et al. (2021). While reward engineering and decomposition is a difficult task,

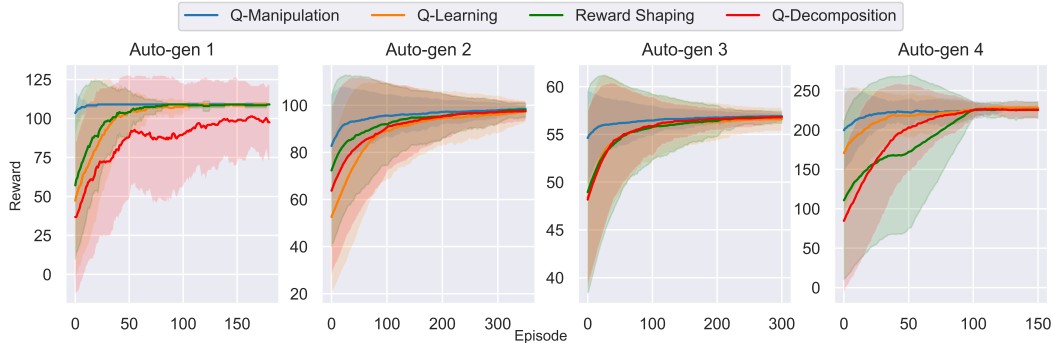

Figure 3: Convergence comparisons for the synthetic domains where $\mathcal{R}$ has a linear form.

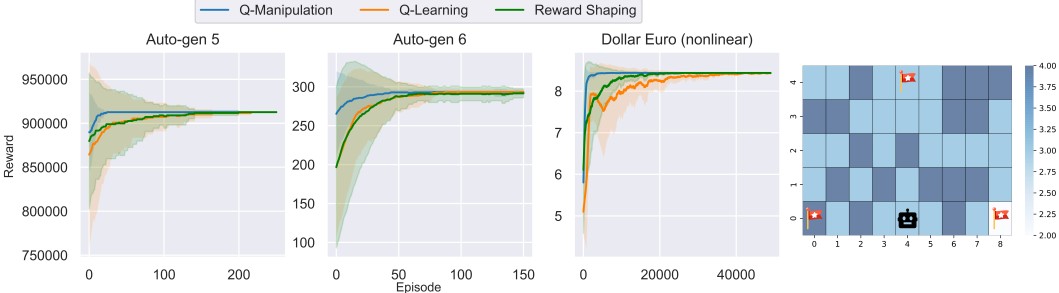

Figure 4: Convergence comparisons and a heat map (Dollar-Euro) for domains where $\mathcal{R}$ has a non-linear form. Q-D is not applicable here due to the requirement of a linear form.

there are prior approaches Lin et al. (2019), Marthi (2007), Ciardo & Trivedi (1993) suggesting novel ways to exploit reward structure and decompose the reward function to better learn. For example, Q-Decomposition Russell & Zimdars (2003) studied a similar problem as ours. It aims to learn a behavior under a reward function that is the linear sum of multiple sub-reward functions. Each sub-agent for such a sub-reward function undergoes its own learning process and supplies its Q values to an aggregator. Q-Decomposition works only with linear sums. The idea has also been extended to work with Deep Q Networks (DQN) Van Seijen et al. (2017). There, it is argued that reward decomposition enables faster learning as separate value functions only depend on a subset of input features, resulting in simpler domains. Similar ideas have been developed in Sutton et al. (2011), Sprague & Ballard (2003), etc. While these ideas are inspirational to our work, they are akin to learning from scratch. Model reuse and modular design are not the focus there.

**Multi-Objective Reinforcement Learning**: Multi-Objective Reinforcement Learning (MORL) Liu et al. (2014), Sprague & Ballard (2003), Roijers et al. (2013), Vamplew et al. (2011) is a branch of RL that deals with learning trade-offs between multiple objectives. A common approach to MORL is to search for the Pareto frontier. A simple way to combine the objectives uses linear scalarization (Van Moffaert et al. (2013)). Often, the domain expert decides

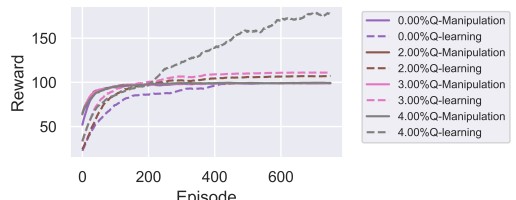

Figure 5: Convergence with different noise levels in target reward function.

the weights for the objectives. Limitations (Vamplew et al. (2008)) have been reported and solutions to counter them include using the Chebyshev function. Our approach can be considered as a special case of MORL where the different objectives can be combined in complex ways. Furthermore, our approach focuses at improving sample complexity via action pruning by utilizing the existing behaviors learned a priori for the individual objectives.

**Hierarchical Reinforcement Learning**: Hierarchical RL (HRL) Dieterich (1998), Vezhnevets et al. (2017), Barreto et al. (2020), Bacon et al. (2017), Barto & Mahadevan (2003), Xiaoqin et al. (2009), Cai et al. (2013), Doroodgar & Nejat (2010) is the process of learning based on a hierarchy of

behaviors, which is often assumed to be known or learned. A hierarchical structure makes it possible to divide a learning problem into sub-problems, sometimes in a recursive manner. At any point in time, a hierarchy of behaviors may be activated and the behavior at the lowest level determines the output behavior. In HRL, the interaction between the behaviors is often assumed to be simple, i.e., sequential execution. In contrast, the interaction between the target behavior and source behaviors in our work can be arbitrarily complex through the correlations between their reward functions. In this aspect, our work contributes a novel perspective to model reuse.

**Transfer Learning and Multi-Task Deep Reinforcement Learning**: Transfer learning is the process of learning a target task by leveraging experiences from source tasks. For example, AlphaGo Silver et al. (2016) uses this technique to learn playing Go from other games. Transfer learning is also applied to natural language processing Andreas et al. (2016); Bahdanau et al. (2016); Chang et al. (2015). As a transfer learning method for reinforcement learning, multi-task reinforcement learning (Vithayathil Varghese & Mahmoud (2020)) deals with learning from multiple related tasks simultaneously to expedite learning. At regular intervals, individual learning agents learn from a related task and share (D'Eramo et al. (2019)) their weights with the global network. The global network also periodically shares its parameters with individual learning agents. Multi-task learning focuses on the combination of network parameters to share the knowledge learned. Our approach also deals with knowledge transfer from the source to the target domains: Q values in the source domains are used to prune actions in the target domain to guarantee better sample complexity. In this regard, it represents the class of indirect transfer methods since the agent must "infer useful knowledge" from the source behaviors before using it.

**Offline Reinforcement Learning**: In contrast to traditional RL methods, offline RL Levine et al. (2020) aims to learn decision-making strategies from offline data without any online interaction. As a result, offline RL can leverage the large amount of training data collected offline. In comparison with Q-M, offline RL is based on the similar ideology of information reuse with a focus on data instead of the previously trained models. One of the challenges that offline RL must deal with is distribution shift. Results from there may be leveraged by Q-M when learning the Q functions (to be maintained for future uses) for the source domains to achieve sample reuse.

## 5 LIMITATIONS & CONCLUSIONS

Our proposed approach to reward adaptation (Q-Manipulation) is limited in many aspects and opens up numerous future opportunities. First, linear programming formulation is a bottleneck. As the numbers of states and actions increase, the number of linear constraints also increases and the approach would quickly become infeasible. We will need to consider approximation methods (such as using abstractions) that still guarantee optimality. In a similar vein, extending Q-Manipulation to continuous domains poses a significant challenge.

Our approach requires both $Q^*$ and $Q^\mu$ to be maintained for the source behaviors. Learning $Q^\mu$ would increase the learning cost even though source domains are assumed to be less expensive while accessible. It would be interesting to see how sample reuse can be achieved to reduce the cost. This is especially useful for including the learned target behavior into the repository of source behaviors for learning other target behaviors. For further improvement, we can consider whether action pruning can be achieved given only $Q^*$ and $\pi^*$. Our work assumes that the target reward function is an exact polynomial function of the source reward functions. In the case this information is unknown or we would only be able to use the available source functions to "approximate" the target function, it would be interesting to study the effectiveness of such approximations (see Sec. 3 for an initial study). A more ambitious direction would be to relax the assumption of the shared state and action spaces to allow arbitrary different source behaviors to be leveraged when learning a target behavior.

In this paper, we introduced reward adaptation, the problem where the learning agent adapted to a target reward function based on the existing source behaviors. Under a few assumptions, we proposed an approach to reward adaptation, referred as Q-Manipulation. The key was to maintain two different Q functions for each of the source behaviors and use them with reward shaping for action pruning before learning the target behavior. We formally proved that our approach retained optimality and thus guaranteed better sample complexity. Empirically, we showed that Q-Manipulation was substantially more efficient than the baselines, generalizable to a variety of domains with different forms of the target reward function. As such, our approach to reward adaptation represents a valuable contribution to advancing reinforcement learning.

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

# A  APPENDIX

## A.1  LEMMA 1:

$$
\begin{aligned}
Q_R^\mu(s, a) &= \min_\pi \left[ \mathbb{E} \left[ \sum_{t=0}^\infty \gamma^t r_t | s_0 = s, a_0 = a, \pi \right] \right] \\
&= -\max_\pi \left[ \mathbb{E} \left[ \sum_{t=0}^\infty -\gamma^t r_t | s_0 = s, a_0 = a, \pi \right] \right] \\
&= -Q_{-R}^*(s, a)
\end{aligned}
\tag{16}
$$

## A.2  DERIVATION FOR BOUNDS OF DIFFERENT FORMS OF $\mathcal{R}$

**Lemma 2:** When $\mathcal{R} = R^m$ and $R$ is positive, an upper and lower bounds of the expected return under $\mathcal{R}$ are given, respectively, by:

$$
\mathbb{Q}_\mathcal{R}^* = Q_R^{*\,m} \geq Q_{R^m}^* = Q_\mathcal{R}^*
\tag{17}
$$

$$
\mathbb{Q}_\mathcal{R}^\mu = -|Q_{-R}^*|^m \leq -Q_{-R^m}^* = Q_\mathcal{R}^\mu
\tag{18}
$$

*Proof.* Ignoring discounting, $Q_R^*$ is specified as:

$$
Q_R^* = \max_\pi \left[ \mathbb{E} \left[ r_0 + r_1 + \ldots + r_n | s_0 = s, a_0 = a, \pi \right] \right]
\tag{19}
$$

and

$$
Q_{R^m}^* = \max_\pi \left[ \mathbb{E} \left[ r_0^m + r_1^m + \ldots + r_n^m | s_0 = s, a_0 = a, \pi \right] \right]
\tag{20}
$$

Denote the optimal policy that corresponds to $Q_{R^m}^*$ as $\pi_m^*$. We know that $Q_R^* \geq Q_R(\pi_m^*)$ (i.e., the Q function under reward $R$ and policy $\pi_m^*$). Given that $R$ is positive, we have that

$$
(r_0 + r_1 + \ldots + r_n)^m \geq r_0^m + r_1^m + \ldots + r_n^m
\tag{21}
$$

Given that $Q_R(\pi_m^*)$ and $Q_{R^m}^*$ follow the same policy under the same domain, we know that

$$
Q_R(\pi_m^*)^m \geq Q_{R^m}^*
\tag{22}
$$

As a result, we have that

$$
(Q_R^*)^m \geq Q_{R^m}^*
\tag{23}
$$

For the lower bound, given that $R$ is positive, we know that $|Q_{-R}^*|^m \geq 0$ and $Q_{-R^m}^* \leq 0$. Hence, $|Q_{-R}^*|^m \geq Q_{-R^m}^*$. $\qquad\square$

**Lemma 3:** When $\mathcal{R} = R_i \times R_j$, and both $R_i$ and $R_j$ are positive, an upper and lower bounds of the expected return under $\mathcal{R}$ is given, respectively, by:

$$
\mathbb{Q}_\mathcal{R}^* = Q_{R_i}^* \times Q_{R_j}^* \geq Q_{R_i R_j}^* = Q_\mathcal{R}^*
\tag{24}
$$

$$
\mathbb{Q}_\mathcal{R}^\mu = -|Q_{-R_i}^*| \times |Q_{-R_j}^*| \leq -Q_{-R_i R_j}^* = Q_\mathcal{R}^\mu
\tag{25}
$$

*Proof.* We know the following holds with positive rewards:

$$
\sum r_i \times \sum r_j \geq \sum r_i r_j
\tag{26}
$$

Denote the optimal policy that corresponds to $Q_{R_i R_j}^*$ as $\pi_{ij}^*$. We know that $Q_{R_i}^* \geq Q_{R_i}(\pi_{ij}^*)$ and $Q_{R_i}^* \geq Q_{R_j}(\pi_{ij}^*)$. Hence, ignoring discounting, we know that

$$
Q_{R_i}^* \times Q_{R_j}^* \geq Q_{R_i}(\pi_{ij}^*) \times Q_{R_j}(\pi_{ij}^*) \geq Q_{R_i R_j}^*
\tag{27}
$$

For the lower bound, given that $R$ is positive, we know that $|Q_{-R_i}^*| \times |Q_{-R_j}^*| \geq 0$ and $Q_{-R_i R_j}^* \leq 0$. Hence, $|Q_{-R_i}^*| \times |Q_{-R_j}^*| \geq Q_{-R_i R_j}^*$.

$\qquad\square$

**Lemma 4:** When $\mathcal{R} = aR_i + bR_j$ $(a > 0, b > 0)$, an upper and lower bounds of the expected return under $\mathcal{R}$ are given, respectively, by:

$$\mathbb{Q}^*_{\mathcal{R}} = aQ^*_{R_i} + bQ^*_{R_j} \geq Q^*_{aR_i+bR_j} = Q^*_{\mathcal{R}} \tag{28}$$

$$\mathbb{Q}^\mu_{\mathcal{R}} = -(aQ^*_{-R_i} + bQ^*_{-R_j}) \leq -Q^*_{-(aR_i+bR_j)} = Q^\mu_{\mathcal{R}} \tag{29}$$

*Proof.* $Q^*_R$ is specified as:

$$Q^*_R = \max_\pi \left[\mathbb{E}\left[r_0 + \gamma r_1 + \ldots + \gamma^n r_n | s_0 = s, a_0 = a, \pi\right]\right] \tag{30}$$

From this, we can derive that

$$Q^*_{aR} = \max_\pi \left[\mathbb{E}\left[ar_0 + \gamma ar_1 + \ldots + \gamma^n ar_n | s_0 = s, a_0 = a, \pi\right]\right]$$
$$= aQ^*_R \tag{31}$$

Denote the optimal policy for $Q^*_{aR_i+bR_j}$ as $\pi^*_{ij}$, given $a > 0$ and $b > 0$, we can derive that

$$aQ^*_{R_i} + bQ^*_{R_j} \geq aQ_{R_i}(\pi^*_{ij}) + bQ_{R_j}(\pi^*_{ij}) = Q^*_{aR_i+bR_j} \tag{32}$$

Since we do not require positive reward above, to derive a lower bound we simply replace $R$ with $-R$ above, and we have

$$(aQ^*_{-R_i} + bQ^*_{-R_j}) \geq Q^*_{-(aR_i+bR_j)} \tag{33}$$

which can then be used to derive the lower bound by multiplying both sides by $-1$. $\qquad\square$

**Theorem 1:** Given $\mathcal{R}$ in the form of Eq. 4 where every $a_{i_1 i_2 \ldots i_n} > 0$ and every source reward function is positive, $\mathbb{Q}^*_{\mathcal{R}}$ and $\mathbb{Q}^\mu_{\mathcal{R}}$ can be derived.

*Proof:* We start with deriving the upper bound for $a_{i_1 i_2 \ldots i_n} R_1^{i_1} \times R_2^{i_2} \ldots R_n^{i_n}$. Let $R_k^{i_k} = \bar{R}_k$, using Lemma 3 we can obtain the following:

$$Q^*_{a_{i_1 i_2 \ldots i_n} \bar{R}_1 \times \bar{R}_2 \ldots \bar{R}_n} \leq a_{i_1 i_2 \ldots i_n} |Q^*_{\bar{R}_1}| \times |Q^*_{\bar{R}_2}| \times \ldots \times |Q^*_{\bar{R}_n}| \qquad \text{[Lemma 3]} \tag{34}$$

Substituting $R_k^{i_k} = \bar{R}_k$ in Eq. 34, we obtain

$$Q^*_{a_{i_1 i_2 \ldots i_n} R_1^{i_1} \times R_2^{i_2} \ldots R_n^{i_n}} \leq a_{i_1 i_2 \ldots i_n} |Q^*_{R_1}|^{i_1} \times |Q^*_{R_2}|^{i_2} \times \ldots \times |Q^*_{R_n}|^{i_n} \qquad \text{[Lemma 2]} \tag{35}$$

Now we can derive the following for $\mathcal{R} = \sum_{i_1+i_2+i_3+\ldots i_n \leq m} a_{i_1 i_2 \ldots i_n} R_1^{i_1} R_2^{i_2} \cdots R_n^{i_n}$:

$$Q^*_{\mathcal{R}} \leq \sum_{i_1+i_2+i_3+\ldots i_n \leq m} a_{i_1 i_2 \ldots i_n} |Q^*_{R_1}|^{i_1} \times |Q^*_{R_2}|^{i_2} \times \ldots \times |Q^*_{R_n}|^{i_n} \qquad \text{[Lemma 4]} \tag{36}$$

Similarly we can derive the lower bound:

$$\mathbb{Q}^\mu_{\mathcal{R}} = - \left[\sum_{i_1+i_2+i_3+\ldots i_n \leq m} a_{i_1 i_2 \ldots i_n} |Q^*_{-R_1}|^{i_1} \times |Q^*_{-R_2}|^{i_2} \times \ldots \times |Q^*_{-R_n}|^{i_n}\right] \leq Q^\mu_{\mathcal{R}} \tag{37}$$

## A.3 ALGORITHM

---

**Algorithm 1** Reward Adaptation Via Q-Manipulation

---

1: Retrieve $Q^*_{R_i}$ and $Q^*_{-R_i}$ from each source behavior index by $i$
2: Compute $\mathbb{Q}^*_{\mathcal{R}}$ and $\mathbb{Q}^\mu_{\mathcal{R}}$ based on the Q and Q-min's ($Q^*_{R_i}$ and $Q^*_{-R_i}$'s) of the source behaviors
3: Tighten the bounds using linear programming:

$$\min_{\Phi(s)} \sum_{s,a} \mathbb{Q}^*_{\mathcal{R}}(s,a) - \mathbb{Q}^\mu_{\mathcal{R}}(s,a) - 2*\Phi(s)$$

$$\text{s.t.} \ \ \forall s \in S, \forall a \in A$$

$$\mathbb{Q}^*_{\mathcal{R}}(s,a) - \Phi(s) \geq \mathbb{Q}^\mu_{\mathcal{R}}(s,a) + \Phi(s)$$

4: Prune action: if $\mathbb{Q}^\mu_{\mathcal{R}_{\mathcal{F}}}(s,a) \geq \mathbb{Q}^*_{\mathcal{R}_{\mathcal{F}}}(s,\widehat{a})$ under a state $s$, $\widehat{a}$ can be pruned.
5: Perform Q-learning on target reward function $\mathcal{R}$ with the remaining set of actions

---

### A.4 ADDITIONAL RESULTS

In this section, we present additional results with synthetic domains where MDPs are auto-generated. In Tab. 2, we present results for domains generated with larger state and action space sizes with linear $\mathcal{R}$. In Tab. 3, results for domains with non-linear $\mathcal{R}$ are presented where we allow the source rewards to be both positive and negative. Note that this conflicts with the theoretical results and hence may lead to the loss of optimality. In practice, for the domains we tested, Q-M still produced the optimal solutions. Convergence comparisons between Q-M and the baselines under these domains are presented in Figs. 6 and 7, respectively. Similar observations were made. These results demonstrated that Q-M can scale to larger domains and that it is somewhat robust to the assumption concerning positive source reward functions in the non-linear target reward function setting. We will study these further in future work.

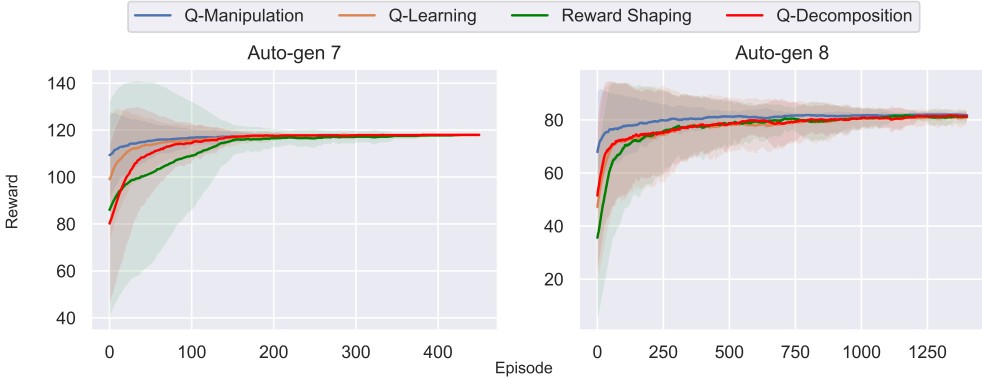

Figure 6: Convergence for larger synthetic domains with linear target reward function

| Environment | $\mathcal{R}$ | $|S|$ | $|A|$ | Actions Pruned | Time (s) | | | |
|---|---|---|---|---|---|---|---|---|
| | | | | | Q | Q-D | Q-M | R-S |
| Auto-gen 7 | $R_1 + R_2$ | 92 | 15 | 288 | 1.43 | 2.71 | 1.19 | 2.04 |
| Auto-gen 8 | $R_1 + R_2$ | 239 | 22 | 1201 | 24.86 | 51.99 | 19.80 | 27.71 |

Table 2: Summary of additional domains of larger state and action space sizes, actions pruned, and time performance comparisons. $\mathcal{R}$ is linear.

| Environment | $\mathcal{R}$ | $|S|$ | $|A|$ | Actions Pruned | Time (s) | | | |
|---|---|---|---|---|---|---|---|---|
| | | | | | Q | Q-D | Q-M | R-S |
| Auto-gen 9 | $R_1 \times R_2 + R_3 + R_4$ | 64 | 18 | 15 | 5.54 | - | 2.18 | 5.19 |
| Auto-gen 10 | $R_1^3 + R_2^3$ | 239 | 22 | 5018 | 21.98 | - | 52.22 | 28.08 |
| Auto-gen 11 | $R_1^3 + R_2^3$ | 19 | 5 | 13 | 1.08 | - | 1.23 | 1.24 |
| Auto-gen 12 | $R_1 \times R_2 + R_3 + R_4$ | 58 | 6 | 247 | 1.27 | - | 1.63 | 1.67 |

Table 3: Summary of additional domains (including domains of larger state and action space sizes), actions pruned, and time performance comparisons. $\mathcal{R}$ is non-linear. Source reward functions can have both positive and negative rewards.

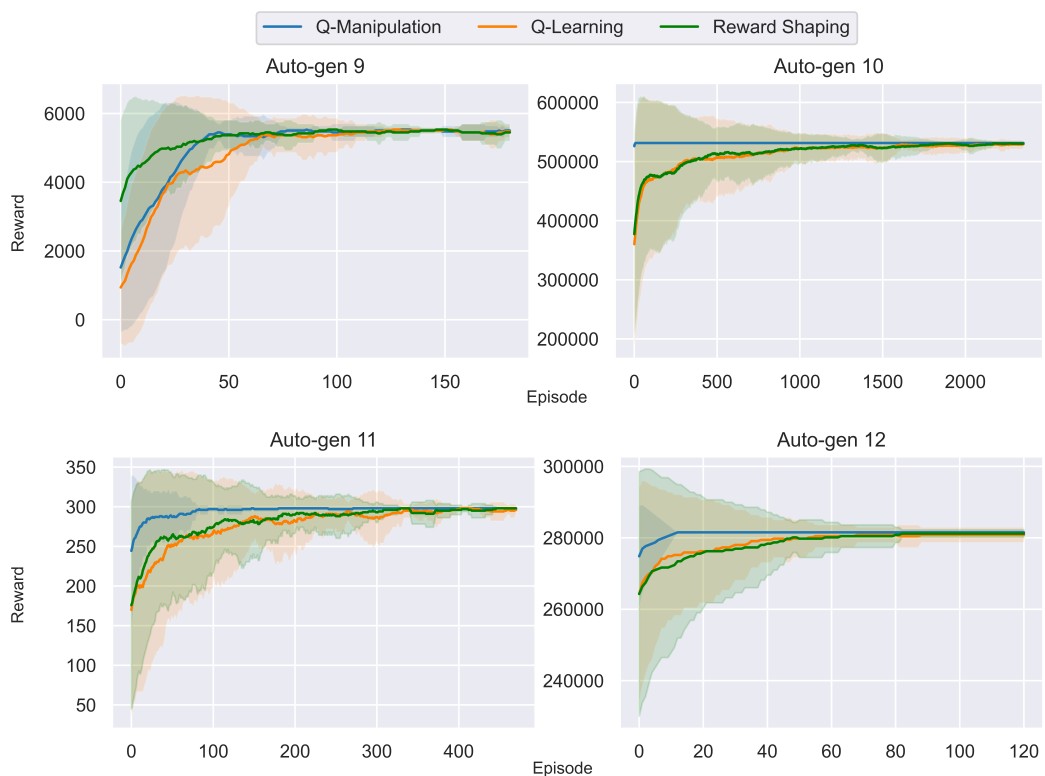

Figure 7: Convergence for synthetic domains with non-linear target reward function with both positive and negative rewards

## A.5 RELATIONSHIP BETWEEN Q-DECOMPOSITION AND SARSA

It is necessary to note that when the individual behaviors use the same training episodes Q-decomposition becomes equivalent to SARSA:

$$\sum_{j=1}^{n} Q_j\left(s_t, a_t\right) \leftarrow \sum_{j=1}^{n}\left(1-\alpha_j^{(t)}\right) Q_j\left(s_t, a_t\right) + \sum_{j=1}^{n} \alpha_j^{(t)}\left[R_j\left(s_t, a_t, s_{t+1}\right) + \gamma Q_j\left(s_{t+1}, \pi(s_{t+1})\right)\right]$$

$$Q\left(s_t, a_t\right) \leftarrow \left(1-\alpha^{(t)}\right) Q\left(s_t, a_t\right) + \alpha^{(t)}\left[R\left(s_t, a_t, s_{t+1}\right) + \gamma Q\left(s_{t+1}, \pi(s_{t+1})\right)\right]$$

Here, Q-decomposition and SARSA both follow the global behavior's policy $\pi(s_{t+1})$ after taking action $a_t$ in state $s_t$. D-Decomposition does not have any guarantees for performance improvement over traditional approaches in terms of sample complexity but has been demonstrated to have limited positive effects. Also, its application is restricted to linear decomposition of the reward function.

