# OpenReview forum: "Reward Adaptation Via Q-Manipulation"
_ICLR.cc/2024/Conference — Submitted to ICLR 2024_

### Official Review · Reviewer_Qwty · 2023-10-30

**Soundness:** 3 good
**Presentation:** 3 good
**Contribution:** 2 fair
**Rating:** 5
**Confidence:** 3

**Summary:**

This paper proposes a novel training schema called "reward adaptation" to leverage existing learned Q functions (with pre-defined source reward settings) to expedite learning a target Q function(in a new and target reward setting). The key idea is to maintain two Q-function variants (Q* and Q-min) for each source reward setting and use them to compute bounds on the expected return under the target reward function. Using reward shaping to tighten the bounds, many actions can be safely pruned before learning the target Q function. This "Q-Manipulation" approach is proven to retain optimality. Empirical results in a variety of domains show faster convergence compared to baselines.

**Strengths:**

1) The problem of reward adaptation provides a new perspective on transfer learning and modular RL. Being able to leverage existing behaviors to learn new ones more efficiently has many useful applications.

2) Theoretical analysis of computing bounds on the expected return and the effects of reward shaping is thorough. The proof of retained optimality after pruning is important.

3) Empirical results demonstrate significantly faster convergence across different domains, validating the effectiveness of Q-Manipulation for pruning unpromising actions. Comparisons to relevant baselines are adequate.

**Weaknesses:**

1) The linear programming formulation for computing the reward shaping function does not scale well. Approximation methods need to be considered for large state/action spaces.

2) Requiring both Q* and Q-min doubles the learning cost for source behaviors. It would be useful to analyze if pruning is possible with just Q* and π*.

3) The assumption of target reward being an exact polynomial of source rewards is limiting. More analysis on effectiveness when the target can only be approximated is needed.

4) Comparisons to a broader variety of baselines (e.g., transfer learning methods) could be more informative about the relative merits of this approach. More results on complex environments, such as Atari Games, should be better inducted to show that this method is effective.

**Questions:**

1) According to Algorithm 1, Is Q* and Q-min pre-learned in an offline form? If so, prune action with steps 1-4 may be considered to be inefficient.

2) The advantage baseline in A2C seems low-cost; how about considering it for pruning action?

3) When does action pruning occur? In the exploration, in the target Q update, or both. It seems that the prune behavior is conservative.

4) Could you visualize the pruned action space in the training process and compare it to the original action space?

---

> ### Author Response · Authors · 2023-11-22
>
> According to Algorithm 1, Is Q* and Q-min pre-learned in an offline form? If so, prune action with steps 1-4 may be considered to be inefficient.
>
> -Since pre-learned Q-values already exist, we plan to reuse those by pruning actions for the target reward function which saves samples usually wasted on exploration.
>
>
> The advantage baseline in A2C seems low-cost; how about considering it for pruning action?
>
> -We plan to consider such approaches in the future.
>
>
> When does action pruning occur? In the exploration, in the target Q update, or both. It seems that the prune behavior is conservative.
>
> -Action pruning occurs before starting to learn for the target behavior. When applying Q-learning for the target reward function, during exploration only set of non-pruned actions is used which helps in faster convergence.
>
>
> Could you visualize the pruned action space in the training process and compare it to the original action space?
>
> -Learning on original action space is equivalent to traditional Q-learning which is used as one of our baselines and learning after pruning action is our approach which performs better in empirical evaluations.

---

### Official Review · Reviewer_hu2d · 2023-10-31

**Soundness:** 2 fair
**Presentation:** 3 good
**Contribution:** 2 fair
**Rating:** 3
**Confidence:** 4

**Summary:**

In this paper, the authors introduce the reward adaptation (RA) problem where the agent learns to adapt to a _target_ reward function while having access to behaviors learned from multiple _source_ reward functions. The authors then focus on a restricted scenario where the target reward function is a polynomial function of the source reward functions. They introduce a method, Q-Manipulation, for action pruning in learning the target task by employing the optimal Q-function from the source tasks, with the hope that learning will be more efficient with the reduced action set. Q-Manipulation estimates the upper bound and the lower bound of the Q-value for each action from its Q-values from the source tasks. An action is then pruned when its Q-value upper bound is below the Q-value lower bound of another action. The authors also introduce a reward shaping based method to tighten the upper bounds and the lower bounds to further facilitate pruning. Tabular experiments demonstrate the effectiveness of Q-Manipulation at action pruning, and further show accelerated learning on the target task.

**Strengths:**

* This work addresses an important question of how to learn more efficiently on a target task by leveraging knowledge from related source tasks. Progress on this question can further broaden the applicability of reinforcement learning to real-world applications.

* Overall the paper is easy to follow. The problem of RA is well motivated and the main idea behind Q-Manipulation is straightforward.

* The experiments are well-designed to test the key attributes of the Q-Manipulation algorithm, such as the portion of pruned action and the downstream impact on learning efficiency.

**Weaknesses:**

* I am afraid that the theoretical results for reward shaping is incorrect. Specifically, I have doubts regarding the correctness of Lemma 5. When reward shaping is applied, $Q_{R_{F}}^{\*}$ and $Q_{R_{F}}^{\mu}$ should be offset towards the same direction, not the opposite directions. In fact, as the authors note in Eq. (15), Lemma 5 implies that $Q^{\*}$ can be smaller than $Q^{\mu}$ after reward shaping, meaning that the optimal policy changes, which conflicts with the theory built in [3].

* The novelty of the RA problem is questionable. In the second paragraph of the Introduction section, the authors write "In this paper, we introduce Reward Adaptation (RA), ..." However, this problem has been widely studied by existing works such as [1] and [2]. Discussion on the connection to these existing works is missing in the manuscript.

* All experiments were conducted in tabular toy environments and the improvement in learning efficiency is marginal. The authors did not comment on the scalability of Q-Manipulation to the function approximation settings. It would be more convincing if the authors can provide more empirical evidence supporting the effectiveness of Q-Manipulation in larger environments.


## References
[1] Barreto _et al_, Successor features for transfer in reinforcement learning, NeurIPS 2017

[2] Barreto _et al_, The option keyboard: Combining skills in reinforcement learning, NeurIPS 2019

[3] Ng _et al_, Policy invariance under reward transformations: Theory and application to reward shaping, ICML 1999

**Questions:**

* Could the authors address my question regarding the correctness of Lemma 5?
* Could the authors comment on the scalability of Q-Manipulation?
* Could the author clarify the connections to existing works such as GPI [1] and Option Keyboards [2]?

---

> ### Author Response · Authors · 2023-11-22
>
> We appreciate your feedback on Lemma 5. We need stronger constraints on the structure of the MDP where Q* is not greater than Q^{\mu} after shaping and we would provide a more concrete reasoning for our approach in the next iteration of this paper. We would also consider related works pointed out and show a comparison if necessary or provide an explanation of how they differ.

---

### Official Review · Reviewer_xHsM · 2023-11-01

**Soundness:** 2 fair
**Presentation:** 3 good
**Contribution:** 2 fair
**Rating:** 3
**Confidence:** 4

**Summary:**

This paper addresses the reward adaptation problem, where an agent with access to optimal behavior in source MDPs must quickly learn optimal behavior in a target MDP with a new reward function. The authors assume the target reward function is a polynomial function of the source reward functions in a finite MDP setting and propose the "Q-Manipulation" method to enable action pruning before learning the target behavior.

**Strengths:**

The paper is overall well-written.

The proposed method is novel and interesting.

**Weaknesses:**

As a core contribution, the authors claim that “We introduce the problem of reward adaptation”. However, I am not sure that it is valid. It seems to me the reward adaptation formulation in Section 2.1 of this paper is an extension of the “Transfer via successor features” problem of [1], where the authors assumed the target reward function to be the linear combination of source reward functions (see Section 4 of [1]). If this is not the case, please clarify. If this is the case, it is important to refer to the successor features literature, and compare your method with the methods proposed in [1], both conceptually and empirically.

Based on the derivations in Sections 2.2 and 2.3, the action pruning strategy is heavily reliant on the assumption that the target reward function is a polynomial function of the source reward functions. In the settings where this assumption is violated, there is a risk that even the optimal actions in the target MDP are pruned. The transfer learning techniques that use potential-based shaping ideas can safely avoid this optimality issue.

In a single source MDP setting, for the reward adaptation problem, the technique proposed in [2] can be applied. Here, the Q-value function in the source domain can serve as a potential function to shape the reward function in the target domain. In this case, the target reward function does not need to be a polynomial function of the source reward function. In the case of multiple source MDP setting, we can use a weighted combination of the Q-values as a potential function to shape the reward function in the target domain. One can learn better weights for combining via the bi-level optimization framework proposed in [3]. It is important to compare your approach to this transfer via shaping technique, both conceptually and empirically.

References:

[1] Barreto et al. Successor Features for Transfer in Reinforcement Learning. 2017.

[2] Brys et al. Policy Transfer using Reward Shaping. 2015.

[3] Hu et al. Learning to Utilize Shaping Rewards: A New Approach of Reward Shaping. 2020.

**Questions:**

Minor comments:

In Section 2.2, the authors state that the influence of discounting can be safely ignored, e.g., when MDPs with absorbing states are considered. In the proofs of Lemma 2 and 3, the discounting factor is ignored; whereas, in the proof of Lemma 4, it is not ignored. Please formally/explicitly write the type of MDPs considered in the proofs.

Due to high/overlapping variance in the convergence plots, it is not very clear that the proposed method outperforms the current set of baselines. The authors could consider additional presentation of the results (e.g. in a tabular form).

---

> ### Author Response · Authors · 2023-11-22
>
> The reward adaptation problem requires knowing source behaviors and how it is combined whereas the “Transfer via successor features” depends on the feature vector and its weight. That work requires a supervised learning of feature and weights whereas RA only aims to reuse behaviors that exists. In addition to this successor features work is dependent on generalized policy improvement theorem where one action can be chosen from available policies at a timestep t, but fails to learn a new policy which none of the source behaviors can achieve combined for the target. Theoretically, a polynomial combination of rewards would be sufficient to approximate any target reward function given enough degrees of polynomial and that’s the reason for defining our problem where the target is a polynomial combination of rewards.

---

### Official Review · Reviewer_Z39j · 2023-11-01

**Soundness:** 1 poor
**Presentation:** 2 fair
**Contribution:** 2 fair
**Rating:** 3
**Confidence:** 4

**Summary:**

This paper proposes a problem they call "Reward adaptation", where an agent which has been previously trained on a set of different reward functions can be more quickly trained on a new reward function. A method is proposed to compute upper and lower bounds on the Q function for a new reward function, given that the reward function is expressed as a polynomial of the existing reward functions and the system had previously kept the q values and the q values of the negative reward from every state in the environment. These upper and lower bounds are used to eliminate actions during exploration for the new reward.

**Strengths:**

The paper looks at an important problem of transfer learning in RL. The justification for the proposed method in 2.2 seems correct, and it should have few downsides if the domain satisfies the assumptions that make the approach possible.

**Weaknesses:**

Section 2.3 appears to have several errors.  In particular, Lemma5 appears to be wrong. A counter-example would be adding a constant potential to every state, which should not increase the min Q value. This is straightforward to check in the 1-state, 1-action where all policies are the same; the min and max Q-values would be the same even after a potential shaping term was added, contradicting Lemma 5.

I believe the error is on the second line of the derivation of (14). -{R_F} after reward shaping is -R + -F, so once we apply Ng et al.1999 we get a + phi(s) term rather than a negative due to the double negative. There is then another negative remaining outside the square brackets, which makes (14) match (13).

Another hint that this has to be wrong is that the constraint is added to (15) to ensure "the upper bound remains greater than or equal to the lower bound", but that should be mathematically impossible if the theorem was valid.

Another issue with this section is that the Ng. 1999 paper requires SAS rewards, but this paper is written with SA rewards, so the theory does not apply directly as stated.


In addition, the paper could also be made significantly more clear. For instance, rather than defining the minimum achievable reward as Q_{-R}, there is a new symbol introduced (which confusingly includes mu), and then it is immediately pointed out that this is the same as Q_{-R}. It seems like this observation is so straightforward as to not need a Lemma, and the added notation not only uses a lot of space but makes the rest of the paper much more difficult to follow.

Many of the methods are also redefined as acronyms halfway through, in a way that is not self-documenting. It becomes excessively difficult to keep track of the differences between Q-M, RS, Q-D, and Q.

Finally, the method is quite difficult to motivate. The main example they point to is if you had a self-driving car that was trained to "either be fast or safe", you could warm-start it to learn the other, but this is far from how self-driving cars work. Even so, it is hard to imagine how "fast" or "safe" could be expressed as a polynomial of the other, as is required by their method.

**Questions:**

How do you think the method could be extended past polynomial combinations of existing reward functions?

---

> ### Author Response · Authors · 2023-11-22
>
> Theoretically, a polynomial combination of rewards would be sufficient to approximate any target reward function given enough degrees of polynomial. That was part of our motivation to solve this problem. We appreciate your feedback on Lemma 5. As you pointed out, we are indeed using 2 different shaping functions which results into an incorrect lower bound. If we used the lower bound with -phi(s) then the upper and lower bounds shift in the same direction and it wouldn’t help pruning. In our next iteration, we plan to study the use of different shaping functions for such a problem where we can guarantee optimality.

---

### Author Response · Authors · 2023-11-22

We would like to express our sincere gratitude to you for your invaluable time and effort spent in reviewing our Paper. We genuinely appreciate the constructive criticism and suggestions provided by the reviewers. We have carefully considered each of the points raised, and we agree that they will significantly enhance the overall impact and clarity of our research. Instead of providing a full rebuttal at this stage, it is perhaps a better idea to keep your comments in mind while improving our paper for the next round. We would study the related works pointed out by reviewers and show comparisons if necessary as we revise our submission. In addition to this, we would also to thank the reviewers for pointing out the error in Lemma 5. Lastly, we will individually clarify details that help the reviewers understand our approach through individual responses to the questions. We will try to incorporate the feedback in the next iteration of this paper.

---

### Meta-Review · Area_Chair_vQNX · 2023-11-30

**Metareview:**

**Summary**: A method is proposed to compute upper and lower bounds on the Q function for a new reward function, given that the reward function is expressed as a polynomial of the existing reward functions.

**Strengths**: Reviewers appreciated the importance of transfer in RL.

**Weaknesses**: Reviewers noted some important theoretical issues with the paper, which the authors acknowledged in the rebuttal. The reviewers also recommended discussing the relationship with successor representations.

During the discussion period, the authors stated that they would not be revising the paper for ICLR, but instead would plan to resubmit to a future conference.

**Justification For Why Not Higher Score:**

All reviewers voted to reject the paper. The paper had an important theoretical issue.

**Justification For Why Not Lower Score:**

N/A

---

### Decision · Program_Chairs · 2024-01-16

Reject